# Bif-1c Attenuates Viral Proliferation by Regulating Autophagic Flux Blockade Induced by the Rabies Virus CVS-11 Strain in N2a Cells

Pengfei Hou,[a,b] Yidi Guo,[a] Hongli Jin,[a,b] Jingxuan Sun,[a] Yujie Bai,[a] Wujian Li,[a,b] Ling Li,[c] Zengguo Cao,[a,b] Fangfang Wu,[b] Haili Zhang,[a] Yuanyuan Li,[a] Songtao Yang,[b] Xianzhu Xia,[b] Pei Huang,[a] ⓘ Hualei Wang[a]

ᵃKey Laboratory of Zoonosis Research, Ministry of Education, College of Veterinary Medicine, Jilin University, Changchun, China
ᵇChangchun Veterinary Research Institute, Chinese Academy of Agricultural Sciences, Changchun, China
ᶜNational Research Center for Exotic Animal Diseases, China Animal Health and Epidemiology Center, Qingdao, China

Pengfei Hou, Yidi Guo, and Hongli Jin contributed equally to this work. Author order was determined by drawing straws.

**ABSTRACT** Bax-interacting factor-1 (Bif-1) is a multifunctional protein involved in apoptosis, autophagy, and mitochondrial morphology. However, the associations between Bif-1 and viruses are poorly understood. As discrete Bif-1 isoforms are selectively expressed and exert corresponding effects, we evaluated the effects of neuron-specific/ubiquitous Bif-1 isoforms on rabies virus (RABV) proliferation. First, infection with the RABV CVS-11 strain significantly altered Bif-1 expression in mouse neuroblastoma (N2a) cells, and Bif-1 knockdown in turn promoted RABV replication. Overexpression of neuron-specific Bif-1 isoforms (Bif-1b/c/e) suppressed RABV replication. Moreover, our study showed that Bif-1c colocalized with LC3 and partially alleviated the incomplete autophagic flux induced by RABV. Taken together, our data reveal that neuron-specific Bif-1 isoforms impair the RABV replication process by abolishing autophagosome accumulation and blocking autophagic flux induced by the RABV CVS-11 strain in N2a cells.

**IMPORTANCE** Autophagy can be triggered by viral infection and replication. Autophagosomes are generated and affect RABV replication, which differs by viral strain and infected cell type. Bax-interacting factor-1 (Bif-1) mainly has a proapoptotic function but is also involved in autophagosome formation. However, the association between Bif-1-involved autophagy and RABV infection remains unclear. In this study, our data reveal that a neuron-specific Bif-1 isoform, Bif-1c, impaired viral replication by unchoking autophagosome accumulation induced by RABV in N2a cells to a certain extent. Our study reveals for the first time that Bif-1 is involved in modulating autophagic flux and plays a crucial role in RABV replication, establishing Bif-1 as a potential therapeutic target for rabies.

**KEYWORDS** Bif-1c, RABV, autophagy flux, replication

Rabies virus (RABV) is a member of the *Rhabdoviridae* family, *Lyssavirus* genus, and *Rabies lyssavirus* species. RABV has a bullet-shaped structure and a single negative-sense RNA genome of approximately 12 kb, which encodes 5 structural proteins: the nucleoprotein (N), phosphoprotein (P), matrix protein (M), glycoprotein (G), and viral RNA polymerase (L). RABV infection causes fatal acute encephalomyelitis in the central nervous system (CNS) and leads to a large number of deaths worldwide every year (1–4). Although many efforts have been made to elucidate the molecular mechanisms of the RABV life cycle, the exact mechanisms underlying RABV pathogenicity remain poorly understood.

Autophagy is a vitally important intracellular catabolic process for the clearance of aberrant cytoplasmic components, both endogenous and exogenous, via their transport

Address correspondence to Hualei Wang, wanghualei@jlu.edu.cn, or Pei Huang, huangpei@jlu.edu.cn.

The authors declare no conflict of interest.

to lysosomes and is involved in a wide variety of physiological and pathophysiological processes, including development, aging, neurodegeneration, cancer, and infectious diseases (5–8). Autophagy is accomplished through six closely connected steps, initiation, nucleation, elongation, closure, maturation, and degradation or extrusion, and is regulated by a complex network composed of various associated proteins (9, 10). Autophagic degradation can be triggered by viral infection and replication (11). Initially, autophagy was considered an important pathway for the degradation and disposal of invading pathogens, such as viruses. However, accumulating evidence has revealed that some viruses can exploit the membrane-bound, protected environment of autophagosomes to generate progeny virions (12, 13). RABV P has been reported to bind to BECN1/Beclin1, activate the CASP2-AMPK-MAPK1/3/11-AKT1-MTOR pathway, and induce incomplete autophagy, during which autophagosomes do not efficiently fuse with lysosomes to benefit RABV replication (14, 15). Interestingly, the induction of autophagic flux by RABV differs by viral strain and infected cell type (16, 17). For example, the wild-type RABV GD-SH-01 strain triggers complete autophagy in human neuroblastoma cells but attenuates autophagic flux in mouse neuroblastoma cells (N2a).

Bax-interacting factor-1 (Bif-1), also called endophilin B1 or SH3GLB1, belongs to the endophilin protein family and is a multifunctional protein involved in apoptosis, autophagy, and mitochondrial morphology (18, 19). Bif-1 binds and activates Bax to perform a proapoptotic function (20). Previous studies have shown that Bif-1 also interacts with Beclin 1 (an autophagy-related protein necessary for autophagosome biogenesis) through UV radiation resistance-associated gene (UVRAG), regulates phosphatidylinositol 3-kinase catalytic subunit type 3 (PI3KC3) activity, and is involved in autophagosome formation (21). Bif-1 localizes with microtubule-associated protein light chain 3 (LC3), autophagy-related gene 5 (ATG5), and autophagy-related gene 9 (ATG9) at the isolation membrane or phagophore and possibly regulates vesicle nucleation by inducing membrane curvature through its N-BAR domain (21). Bif-1 is selectively translated into different isoforms in a cell type-specific manner. Neurons selectively express the longer alternatively spliced Bif-1 isoforms, which play a neuroprotective role by attenuating apoptosis and promoting neuronal survival, in sharp contrast to the proapoptotic role previously documented in nonneuronal cells (22). Moreover, autophagy can abolish apoptosis during RABV infection (16). These previous results suggested that Bif-1 may be involved in autophagy regulation and act differently according to the cell-specific spliced isoforms of different lengths. We found that Bif-1 expression was regulated in RABV-infected cells. To explore the potential roles of Bif-1 in RABV infection, we conducted experiments using the CVS-11 strain. The efficient induction of incomplete autophagy by CVS-11 was confirmed in N2a cells. In addition, we found that overexpression of neuron-specific Bif-1 isoforms alleviated the autophagic flux blockade induced by the RABV CVS-11 strain in N2a cells, resulting in decreased viral replication. Our study showed for the first time that Bif-1c is involved in modulating autophagic flux and plays crucial roles in RABV replication.

## RESULTS

**CVS-11 infection increases Bif-1 expression.** To determine whether Bif-1 is differentially expressed in neuronal cells after RABV infection, N2a cells were infected with the CVS-11 strain. Lysates from N2a cells were harvested every 12 h during CVS-11 infection, and the expression of Bif-1 was detected by quantitative RT-PCR (qRT-PCR) and Western blot. The mRNA level of Bif-1 increased in a time-dependent manner, peaking at 48 h postinfection (hpi) in N2a cells compared with mock-infected cells (Fig. 1A). However, the protein level of Bif-1 was elevated at 24 hpi but decreased at later stages of infection, a course that was not identical to changes in its transcription level (Fig. 1B and C). These data revealed that CVS-11 infection upregulates Bif-1 expression and suggested that Bif-1 may in turn affect RABV replication in N2a cells.

**Knockdown of Bif-1 expression promotes CVS-11 replication.** To investigate whether Bif-1 influences RABV replication, we knocked down Bif-1 expression by transfecting cells with shRNA targeting Bif-1 before RABV infection and measured the copy

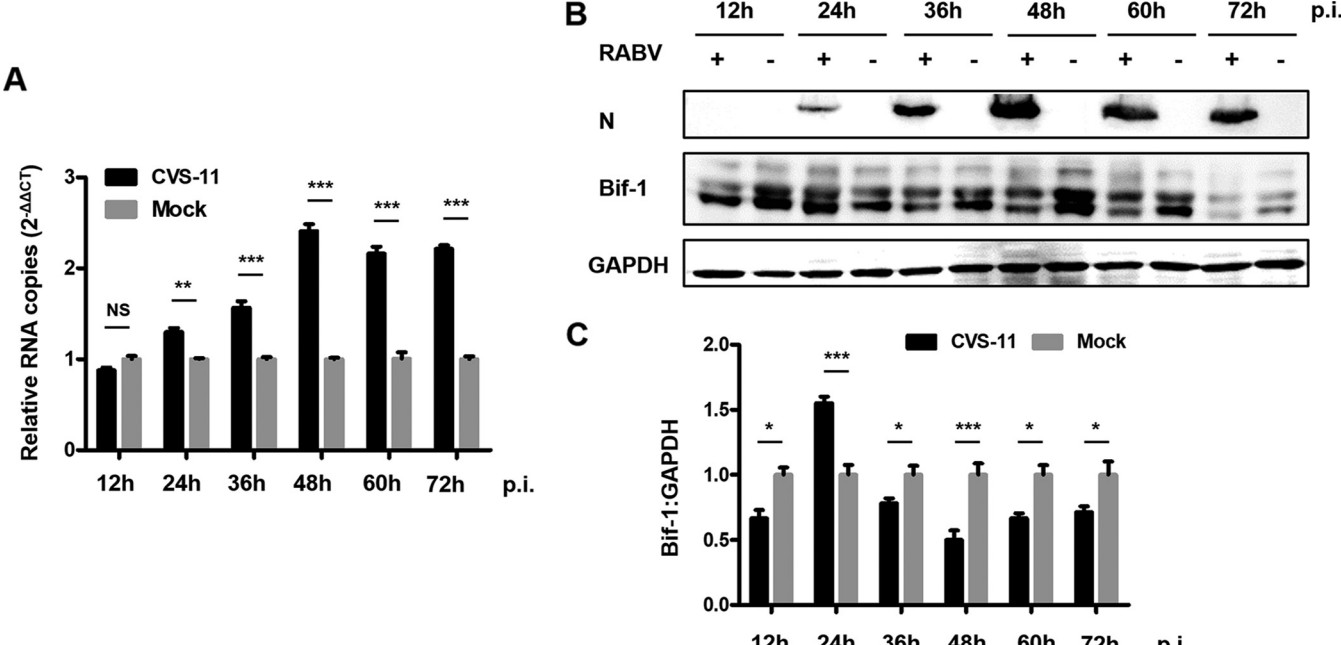

**FIG 1** RABV infection upregulates the expression of Bif-1 in N2a cells. N2a cells were mock-infected or infected with CVS-11 at an MOI of 5. Cell lysates were collected at 12, 24, 36, 48, 60, and 72 hpi. The RNA copy numbers of Bif-1 were determined by qRT-PCR (A), and protein expression levels were determined by Western blot analysis (B). The relative protein level of Bif-1 was analyzed using ImageJ. GAPDH was used as the loading control (C). The results are presented as the mean $\pm$ SD of three independent experiments. The statistical significance of the differences is indicated. Student's $t$ test; $P < 0.05$ (*); $P < 0.01$ (**); $P < 0.001$ (***).

number of RABV genomic RNA and the titer of progeny virus at 24 and 48 hpi. The qRT–PCR results showed that Bif-1 knockdown caused an evident increase in RABV genomic RNA at 24 hpi, while no significant difference was observed at 48 hpi in N2a cells compared to mock-infected cells (Fig. 2A). An increase in the virus titer in the cell culture supernatant by Bif-1 knockdown was validated at 24 hpi by a 50% tissue culture infective dose (TCID$_{50}$) assay (Fig. 2B). There was no significant change in the virus titer at 48 hpi. These results suggested that knockdown of Bif-1 expression promotes CVS-11 infection, as indicated by both the mRNA transcript level and virus titer.

**Overexpression of neuron-specific Bif-1 isoforms impairs CVS-11 replication in N2a cells.** Given that Bif-1 isoforms are selectively expressed in neurons and nonneuronal cells and have contrasting effects on apoptosis (22), we hypothesized that different Bif-1 splice isoforms have distinct functions in RABV infection. To study how Bif-1 iso-

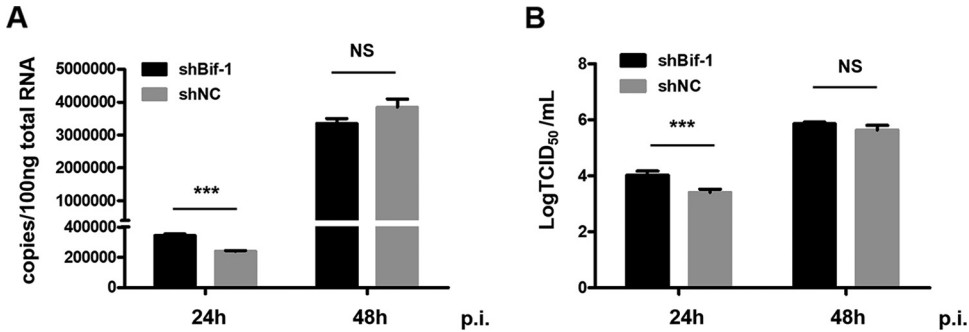

**FIG 2** Knockdown of Bif-1 expression increases RABV infection. N2a cells were transfected with shNC or shBif-1 for 24 h and were then infected with CVS-11 at an MOI of 0.1. Cell lysates were collected at 24 and 48 hpi. The RNA copy numbers of RABV N were determined by qPCR (A). Cell culture supernatants were collected at 24 and 48 hpi. Virus titers were calculated (B). shNC, nontargeting shRNA, negative control. The results are presented as the mean $\pm$ SD of three independent experiments. The statistical significance of the differences is indicated. Student's $t$ test; $P < 0.05$ (*); $P < 0.01$ (**); $P < 0.001$ (***).

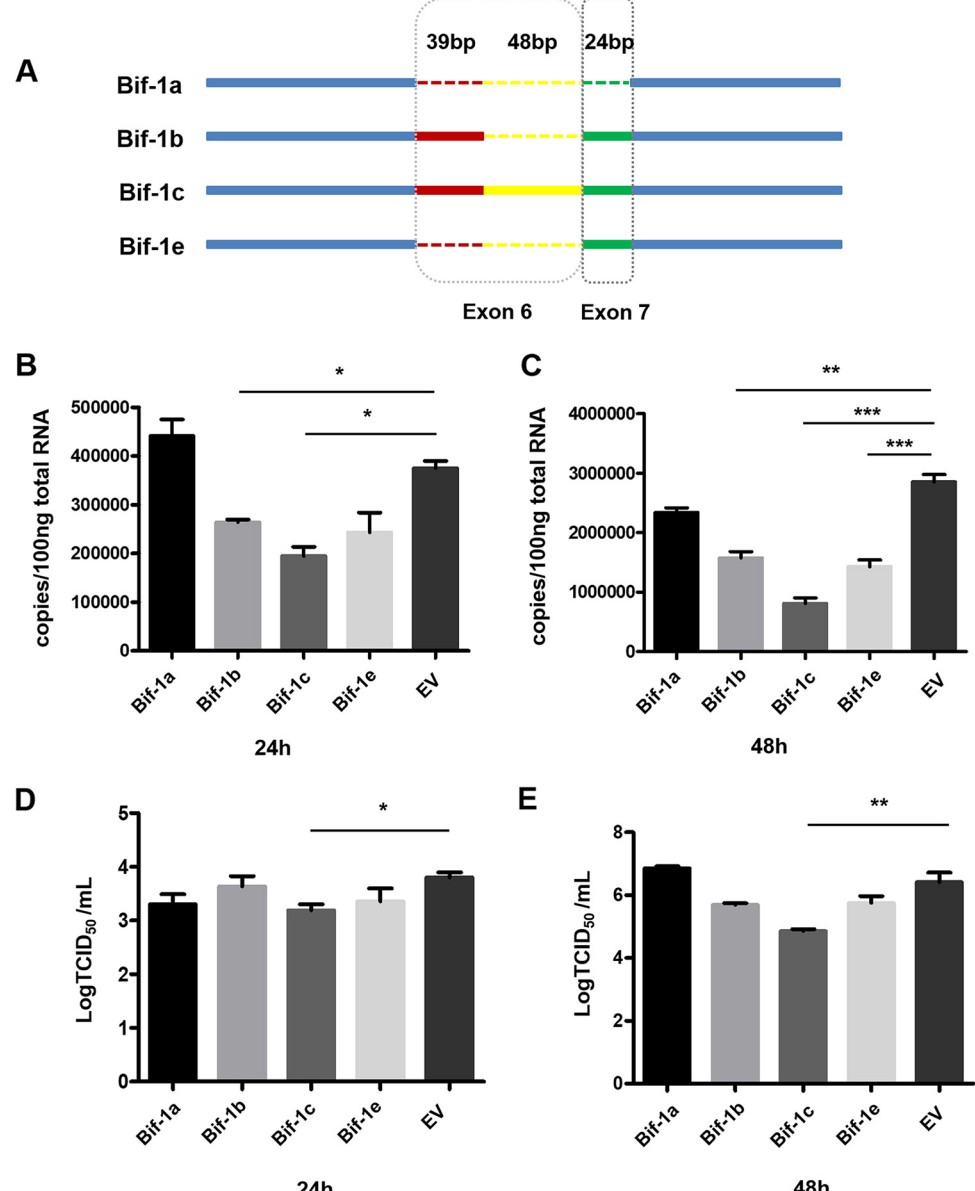

**FIG 3** Overexpression of neuron-specific Bif-1 isoforms attenuates RABV infection in N2a cells. (A) Schematic showing the structure of exon 6 and exon 7 in Bif-1 isoforms. N2a cells transfected with Bif-1a, Bif-1b, Bif-1c, Bif-1e or EV were infected with CVS-11 at an MOI of 0.1. The RNA copy numbers of RABV N were determined by qPCR analysis at 24 hpi (B) and 48 hpi (C). $TCID_{50}$ values were calculated at 24 hpi (D) and 48 hpi (E). The results are presented as the mean $\pm$ SD of three independent experiments. The statistical significance of the differences is indicated. Student's $t$ test; $P < 0.05$ (*); $P < 0.01$ (**); $P < 0.001$ (***). EV, empty vector.

forms are involved in the regulation of viral infection in neuronal cells, we constructed 4 plasmids encoding isoforms of Bif-1 of different lengths (Fig. 3A). Bif-1a lacks exons 6 and 7 and is ubiquitously expressed. The Bif-1c genome is the longest of the five Bif-1 isoforms mentioned in this study, containing exons 6 and 7, whereas Bif-1c contains a long (87 bp) form of exon 6 rather than the short (39 bp) form contained in Bif-1b. Bif-1e contains exon 7 but lacks exon 6. Only neurons and neuroblastoma cells have been reported to express the longer isoforms of Bif-1 (Bif-1b, Bif-1c, and Bif-1e), while Bif-1a, the shortest isoform, was found to be expressed in all other cell types tested (22). The inclusion/exclusion of exons 6 and 7, located within the N-BAR domain responsible for membrane binding and curvature, accounts for the opposite effects of the isoforms on apoptosis (23). N2a cells were transfected with recombinant plasmids encoding Bif-1a,

Bif-1b, Bif-1c, Bif-1e, and EV, respectively. After 36 h, cell lysates were collected and analyzed by Western blot. Bands corresponding to the expected sizes of Bif-1a, Bif-1b, Bif-1c, and Bif-1e were detected, indicating successful expression of the target proteins (data not shown). As shown in Fig. 3B and C, individual overexpression of the longer Bif-1 isoforms (Bif-1b, Bif-1c, and Bif-1e) decreased the RABV genomic RNA copy number at both 24 hpi and 48 hpi, although the suppressive effect of Bif-1e at 24 hpi was not significant. In contrast, Bif-1a, the shortest isoform, had no effect on CVS-11 replication. The virus titer in the cell culture supernatant was also determined by a $TCID_{50}$ assay. Transfection with Bif-1c reduced viral proliferation at 24 hpi and 48 hpi, but transfection of the other Bif-1 isoforms did not lead to significant changes in the virus titer (Fig. 3D and E). Collectively, these data confirmed that the longer neuronal cell-specific Bif-1 isoforms attenuate CVS-11 infection in N2a cells, while the ubiquitously expressed Bif-1a isoform has no effect.

**CVS-11 infection induces autophagosome accumulation and blocks autophagic flux.** To investigate whether CVS-11 infection induces autophagy, N2a cells were transfected with EGFP-LC3 plasmids and then infected with CVS-11 or treated with Rapa, an autophagy activator. LC3 puncta accumulation was clearly observed in RABV-infected and Rapa-treated cells (Fig. 4A). The number of EGFP-LC3 puncta indicating autophagosomes per cell was increased significantly in the CVS-11-infected and Rapa-treated groups compared with the mock-infected group (Fig. 4B). The induction of autophagy was also validated by Western blot analysis. The protein levels of LC3-I, LC3-II, SQMT1/p62, and RABV N were determined and analyzed by normalization to the GAPDH level at different time points postinfection (Fig. 4C to E). The conversion of LC3-I to LC3-II increased over time and was accompanied by an increased level of p62, a marker of autophagic flux, which correlated with the increased number of LC3 puncta observed by confocal microscopy. Consistent with previous reports (14), these data demonstrated that CVS-11 infection induces autophagosome accumulation but blocks autophagic flux.

**Overexpression of Bif-1 enhances autophagic flux in N2a cells.** Considering the crucial roles of Bif-1 in regulating cell autophagy and because the neuron-specific Bif-1 isoform Bif-1c had the greatest effect on RABV replication, we next examined the mechanism by which Bif-1c functions in RABV-induced autophagy. After transfection with red fluorescent protein-fused pDsRed1-Bif-1c and EGFP-LC3 plasmids for 24 h, N2a cells were infected with CVS-11 or treated with Rapa. Cells were then fixed and subjected to immunoconfocal microscopy to assess the colocalization of pDsRed1-Bif-1c and EGFP-LC3 (Fig. 5). We observed that either CVS-11 infection or Rapa treatment resulted in aggregation of LC3-positive puncta in both empty vector (EV)- and Bif-1c-transfected cells. Moreover, when Bif-1c was overexpressed, the LC3 puncta obviously colocalized with Bif-1, indicating that Bif-1 is located at the isolation membrane with LC3 and is involved in autophagosome formation.

To further investigate the involvement of Bif-1 in autophagic flux, Western blot analysis was employed to determine the expression levels of autophagy-associated proteins. CVS-11 infection enhanced the conversion of LC3-I to LC3-II. When Bif-1c was overexpressed, the protein level of LC3-II decreased, accompanied by significant progressive degradation of p62 (Fig. 6A). Bif-1 forms a complex with BECN1/Beclin 1 and UVRAG and regulates autophagosome-lysosome fusion (21). The expression levels of UVRAG and BECN1 were simultaneously increased with upregulation of Bif-1. The ATG14-binding PI3KC3-BECN1 complex is involved in the formation of autophagosomes. The trend in ATG14 expression was consistent with that in p62 expression, which was almost completely abolished after Bif-1 overexpression. Moreover, the mean band densities of p62, BECN1, UVRAG, and ATG14 were analyzed, as shown in Fig. 6B to F, with normalization to GAPDH. We transfected N2a cells with EGFP-LC3 and labeled lysosomes with LysoTracker Red (50 nM), and live-cell imaging was performed. Increased colocalization of LC3 and lysosomes was found after overexpression of Bif-1c compared with that in mock-infected cells or cells with CVS-11 infection only. Rapa treatment further promoted LC3 puncta accumulation and localization to lysosomes (Fig. 6G). Taken together, these data suggest that overexpression of Bif-1c abolishes the blockade of autophagic flux induced by CVS-11 in N2a cells.

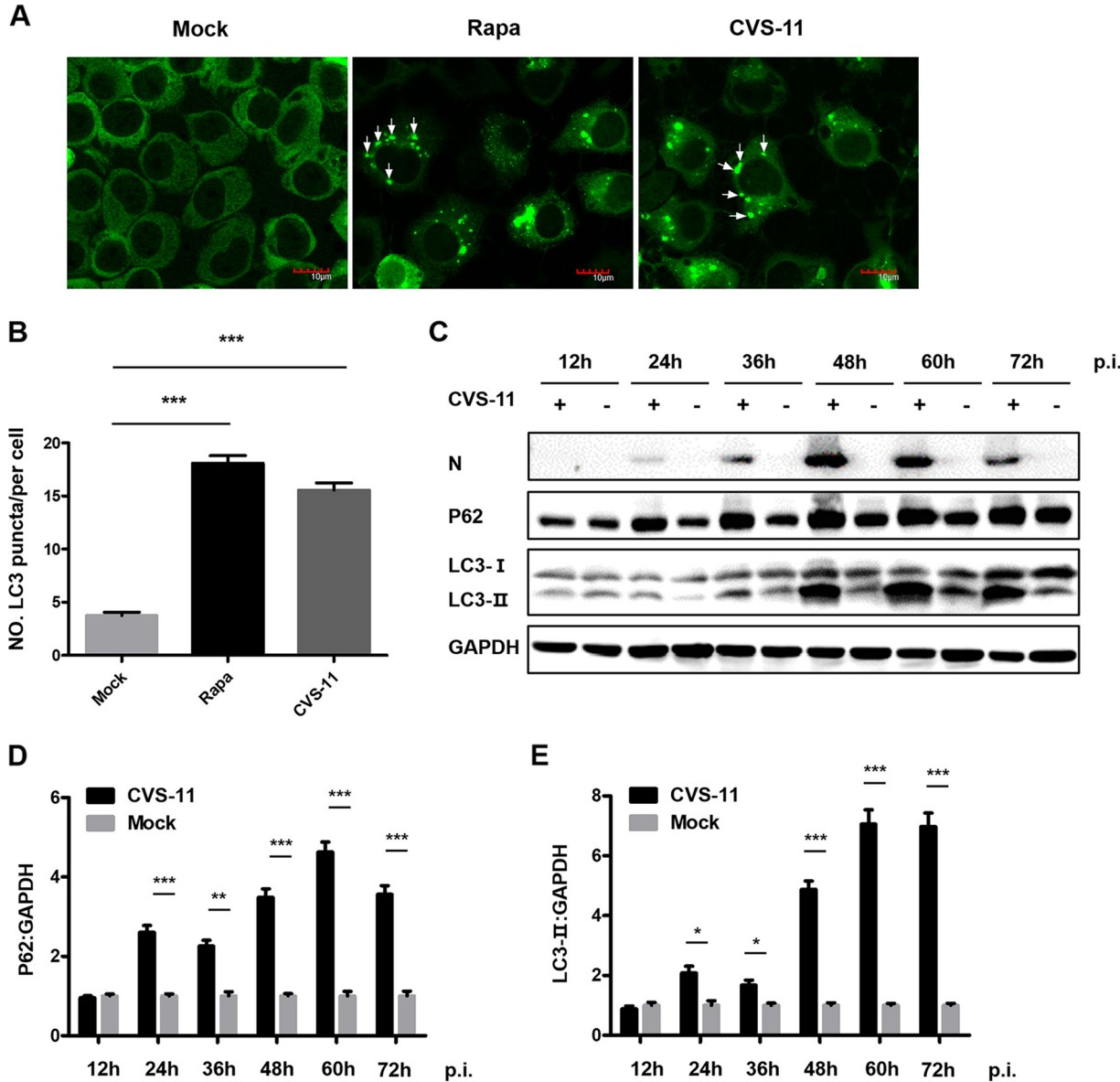

**FIG 4** RABV infection induces autophagy and promotes autophagosome accumulation. (A) N2a cells were transfected with EGFP-LC3 and then infected with CVS-11 at an MOI of 5 or treated with Rapa (1 $\mu$M). After 48 h, the cells were fixed and visualized by confocal microscopy. LC3 puncta are indicated by arrows. Scale bars = 10 $\mu$m. (B) The number of LC3 puncta per EGFP-positive cell was counted (mean $\pm$ SD; $n$ = 35). (C) N2a cells were mock-infected or infected with CVS-11 at an MOI of 5. Cell lysates were collected at 12, 24, 36, 48, 60, and 72 hpi. The expression levels of RABV N, p62, and LC3 were determined by Western blot analysis. The relative protein levels of p62 (D) and LC3-II (E) were analyzed using ImageJ. GAPDH was used as the loading control. The results are presented as the mean $\pm$ SD of three independent experiments. The statistical significance of the differences is indicated. Student's $t$ test; $P < 0.05$ (*); $P < 0.01$ (**); $P < 0.001$ (***).

## DISCUSSION

Bif-1 was originally discovered as a Bax-interacting factor that activates Bax and promotes apoptosis (18). Bif-1 is also required for mitochondrial fission and vesicle formation (24, 25). Moreover, Bif-1 forms a complex with Beclin1 and UVRAG to activate PI3KC3 and induce autophagy in mammalian cells (21, 26). Colocalization of Bif-1 and LC3 at the phagophore membrane has also been observed, implying that Bif-1 is involved in membrane curvature during autophagy. However, few studies have addressed the relationship between Bif-1 and viruses. As RABV is a neurotropic virus, we selected N2a cells for detection of Bif-1 expression after CVS-11 infection. We found a time-dependent increase in the Bif-1 mRNA level (Fig. 1A), which was validated in mouse brain inoculated with CVS-11 (data not shown). Interestingly, the protein level of

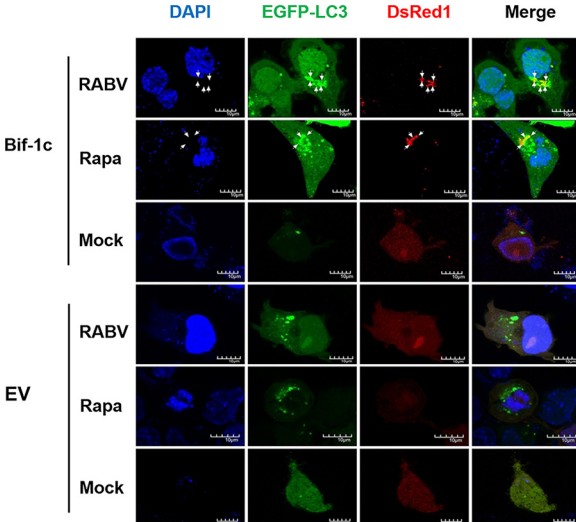

**FIG 5** Bif-1c colocalizes with LC3. N2a cells were transfected with a mixture, including EGFP-LC3 and pDsRed1-Bif-1c, or EV and then infected with CVS-11 at an MOI of 5 or treated with Rapa (1 $\mu$M). At 48 hpi, the cells were fixed and visualized by confocal microscopy. Colocalization of Bif-1 with LC3 puncta is indicated by arrows. DAPI (blue) was used to stain nuclear DNA. Scale bars = 10 $\mu$m. EV, empty vector.

Bif-1 was not entirely consistent with the mRNA level as determined by qRT-PCR, which was elevated at 24 hpi but decreased thereafter (Fig. 1B and C). We speculated that Bif-1 probably underwent autophagic degradation. Additionally, Bif-1 may undergo posttranslational modifications (PTMs), such as ubiquitination, which may affect Bif-1 stability, leading to its proteasomal degradation. Different Bif-1 isoform proteins are differentially distributed in somatic and neuronal cells, Bif-1a, which is shortest, has a proapoptotic function, but the longer spliced isoforms selectively, which are expressed in neuronal cells, have the opposite effect (22). We hypothesized that different Bif-1 isoforms also play distinct roles during RABV infection. As expected, overexpression of Bif-1c, the longest neuron-specific isoform, significantly attenuated viral replication, while shorter Bif-1 isoforms had no significant effect (Fig. 3). Bif-1c was more effective than Bif-1b and Bif-1e, probably because Bif-1c contains the most integrated functional domains. However, in BHK-21 cells, a nonneuronal cell type, we found that Bif-1c barely affected CVS-11 infection, while Bif-1a promoted CVS-11 infection (data not shown). Moreover, knockdown of Bif-1 promoted CVS-11 infection in N2a cells (Fig. 2A and B), and the silencing of nonneuronal spliced Bif-1a inhibited viral replication in BHK-21 cells (data not shown). The roles of Bif-1 in RABV infection in neuronal and nonneuronal cells probably differ. As different RABV strains have different impacts on autophagy (16), their replication may be differentially affected by Bif-1. Given that incomplete autophagy is correlated with apoptosis (27) and that Bif-1 functions to promote apoptosis, we also speculate that Bif-1 simultaneously impacts apoptosis during RABV infection. Whether discrete Bif-1 isoforms exert different effects on apoptosis needs further examination.

Autophagy is a vitally important intracellular catabolic process for the clearance of intracellular pathogens, including bacteria and viruses. Many viruses have been reported to subvert the autophagic machinery to benefit self-replication (12). RABV infection induces incomplete autophagy, as evidenced by the observation that the accumulation of EGFP-LC3 puncta was accompanied by LC3 conversion (Fig. 4). To further explore the mechanism by which Bif-1c regulates RABV infection, we transfected Bif-1c into N2a cells before CVS-11 infection or Rapa treatment, which allowed us to assess the effects of exons 6 and 7, which are responsible for the membrane binding and curvature functions of Bif-1. Consistent with previous findings that Bif-1 and LC3 may cooperate during phagophore membrane formation during autophagy (28), Bif-1 colocalized with LC3 and guided autophagic vacuoles to lysosomes in the later stage of RABV infection (Fig. 5). Moreover, overexpression of Bif-1c alleviated the autophagic flux blockade induced by RABV, as shown

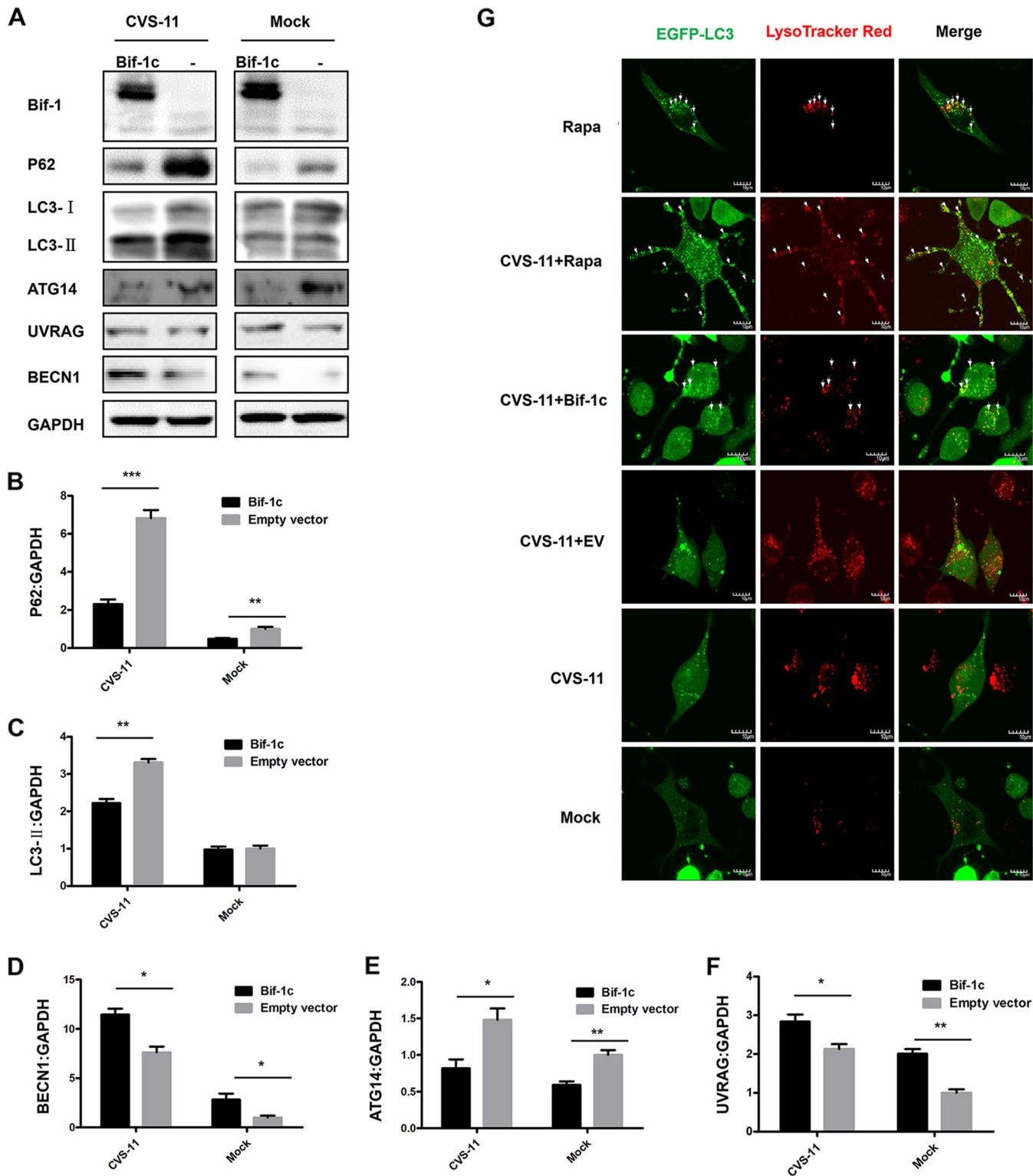

**FIG 6** Bif-1 overexpression promotes autophagosome fusion with lysosomes in RABV-infected cells. N2a cells were transfected with Bif-1c and then infected with CVS-11 at an MOI of 5 or mock-treated. (A) After 48 h, cell lysates were collected, and the expression levels of RABV N, Bif-1, p62, LC3, ATG14, UVRAG, and BECN1 were determined by Western blot analysis. The relative protein levels of p62 (B), LC3 (C), BECN1 (D), ATG14 (E), and UVRAG (F) were analyzed using ImageJ. GAPDH was used as the loading control. (G) N2a cells were transfected with a plasmid mixture including EGFP-LC3 and pcDNA3.1-Bif-1c or EV for 24 h and then infected with CVS-11 at an MOI of 5 and/or treated with Rapa (1 $\mu$M) for 48 h. Cells were incubated with LysoTracker red (50 nM) for 15 min. Live-cell images were obtained by confocal microscopy. Colocalization of EGFP-LC3-labeled autophagosomes (green) with LysoTracker red-stained acidified vesicles (red) is indicated by arrows. EV, empty vector. The results are presented as the mean $\pm$ SD of three independent experiments. Scale bars, 10 $\mu$m. The statistical significance of the differences is indicated. Student's $t$ test; $P < 0.05$ (*); $P < 0.01$ (**); $P < 0.001$ (***).

by a decrease in LC3-II and the degradation of p62 (Fig. 6). Intracellular autophagosome accumulation has been reported to trigger RABV replication (14). We believe that Bif-1 plays a crucial role in autophagy, a self-protection response orchestrated by hosts upon viral infection. RABV infection stimulated Bif-1 expression. Bif-1 promoted autophagosome fusion with lysosomes, abolished the protective environment of autophagosomes hijacked by RABV, and ultimately suppressed viral infection. The viral protein M of the virulent wild-type RABV strain also stimulated the autophagic response but not when transfected alone (16). BECN1 has been shown to bind the viral protein P and activate the AMPK-MTOR and AMPK-MAPK pathways (14, 15). Bif-1 forms a complex with BECN1 and UVRAG, which implies that RABV P may associate with Bif-1 and participate in Bif-1-mediated autophagy modulation. Given that GTPase Rab7 colocalizes with RABV in neuronal and SH-SY5Y cells (29) and is involved in autophagosome-lysosome fusion (30), Rab7 is likely to play a role in RABV-induced autophagy. In summary, our report has shed light on the role of Bif-1 in CVS-11 replication and autophagic flux regulation, but the precise molecular mechanisms by which Bif-1 isoforms exhibit distinct features and the involvement of autophagy-related signaling pathways remain to be examined.

Neuron-specific Bif-1c impairs CVS-11 replication and abolishes the blockade of autophagic flux induced by CVS-11 in N2a cells. Our study not only demonstrates an unexpected role for Bif-1 in regulating autophagic flux but also establishes Bif-1 as a potential therapeutic target for rabies.

## MATERIALS AND METHODS

**Cell lines and virus.** N2a cells were obtained from ATCC, grown in Dulbecco's modified Eagle's medium (DMEM; Corning) supplemented with 10% fetal bovine serum (FBS; BI) and 1% penicillin/streptomycin, and maintained in a humidified incubator at 37°C and 5% $CO_2$.

The standard virus strain (CVS-11) used for challenge was propagated and titrated in baby hamster kidney (BHK) cells (31). Virus titers were determined by calculating the half-maximal tissue culture infectious dose ($TCID_{50}$) using the Reed-Muench method, as previously described (32, 33). The virus stock was subsequently aliquoted and stored at −80°C. Cells were infected with RABV at an MOI of 1. Cells were washed three times with phosphate-buffered saline (PBS; Sigma) prior to culture in fresh medium at 37°C and 5% $CO_2$ for the indicated times.

**Reagents and antibodies.** The reagent rapamycin (Rapa; R0395; Sigma-Aldrich) was used in the study. The primary antibodies used were anti-ATG14 (no. 96752; Cell Signaling Technology), anti-UVRAG (no. 13115; Cell Signaling Technology), anti-BECN1 (no. 54101; Cell Signaling Technology), anti-p62 (P0067; Sigma-Aldrich), anti-LC3 (L8918; Sigma-Aldrich), anti-Bif1 (NBP2-24733; Novus), anti-GAPDH (RM2002; Beijing Ray Antibody Biotech), anti-RABV (5B12) (NB110-7542; Novus), and fluorescein isothiocyanate (FITC)-conjugated anti-RABV (800-092; FUJIREBIO). The secondary antibodies used were horseradish peroxidase (HRP)-conjugated goat antirabbit IgG (ab97051; Abcam) and goat antimouse IgG (sc-2005; Santa Cruz).

**Plasmid construction and transfection.** The genes encoding the mouse Bif-1a and Bif-1e isoforms were PCR-amplified from N2a cell genomic cDNA with gene-specific primers (upstream primer 5′-ATGAACATCATGGATTTCAACGTG-3′ and downstream primer 5′-TTAATTGAGAAGTTCTAAGTAGGTAATTGG-3′) and cloned into the pIRES2-EGFP vector. The Bif-1b gene was amplified from N2a cell genomic cDNA using 2 pairs of specific primers (pair 1: upstream primer 5′-ATGAACATCATGGATTTCAACGTG-3′ and downstream primer: 5′-TTTTGTCACTTCCTCTGCCCAAATCATAATG-3′; pair 2: upstream primer 5′-ATTTGGGC AGAGGAAGTGACAAAATCTG-3′ and downstream primer 5′-TTAATTGAGAAGTTCTAAGTAGGTAATTGG-3′) and cloned into the pIRES2-EGFP vector. The Bif-1c isoform was amplified with 2 pairs of specific primers (pair 1: upstream primer: 5′-ATGAACATCATGGATTTCAACGTG-3′ and downstream primer 5′-TACATG CAGGAAGTTGAGCATGTAAGAGAAATTTACCATAATGTTATCTCCTTCAG-3′; pair 2: upstream primer 5′-TACATGCTCAACTTCCTGCATGTAAAATGGCTGAAGATTTGGGCAGAGGAAGTG-3′ and 5′-TTAATTGAGAA GTTCTAAGTAGGTAATTGG-3′) for the pIRES2-EGFP vector and pDsRed1-N1 vector. Cells were transfected with these plasmids with Lipofectamine 3000 transfection reagent (L3000008; Invitrogen) according to the manufacturer's instructions. At 24 h posttransfection, cells were infected with RABV for the indicated times and harvested for further analysis.

**Short hairpin RNA knockdown.** Short hairpin RNA (shRNA) targeting Bif-1 (shBif-1: 5′-AGGAA TTGAGAATAACTCAAAG-3′) and negative-control shRNA (shNC: 5′-GTTCTCCGAACGTGTCACGT-3′) were purchased from GenePharma. shRNAs were transfected into cells with Lipofectamine 3000 transfection reagent (L3000008; Invitrogen) according to the manufacturer's instructions for 24 h and then infected with RABV. Cells were harvested for further analysis.

**RNA extraction and quantitative real-time reverse transcription-PCR analysis.** RNA was extracted from cells using a SimplyP Total RNA extraction kit (BSC52S1; BioFlux). First-strand cDNA was synthesized using a PrimeScript RT reagent kit with gDNA Eraser (RR047A; TaKaRa). TaqMan real-time RT–PCR assays were performed in an Mx3000P real-time PCR system (Stratagene) using Premix Ex Taq (Probe qPCR) (RR390L; TaKaRa) according to the manufacturer's instructions. Each 25-$\mu$L reaction mixture contained 12.5 $\mu$L of 2× Premix Ex Taq (Probe qPCR), 0.5 $\mu$L of 10 $\mu$M probe, 0.5 $\mu$L

**TABLE 1** Primers used for real-time PCR

| Gene | Primer/probe | Sequence (5′–3′) |
| --- | --- | --- |
| Bif-1 | Forward | GTAGCCTTGTAATCACCTGTCC |
| | Reverse | GTGACAGTTCAGTGCTATTTGC |
| | Probe | CCTAGCCTTCCTGTTGTTGCTGGAT |
| GAPDH | Forward | TCCAGTATGACTCCACTC |
| | Reverse | GACTCCACGACATACTCA |
| | Probe | CGGCAAATTCAACGGCACAGTC |
| RABV-N | Forward | TCAAGAATATGAGGCGGCTG |
| | Reverse | TGGACGGGCTTGATGATTGG |

each of 10 $\mu$M forward and reverse primers, 0.25 $\mu$L of 50× ROXII reference dye, 9.75 $\mu$L of nuclease-free water, and 1 $\mu$L of cDNA template. The thermal cycling conditions were as follows: 1 cycle at 95°C for 30 s, followed by 40 cycles at 95°C for 5 s, and 60°C for 35 s. A SYBR Premix Ex TaqII kit (RR420A; TaKaRa) was also used for RT-qPCR. Each 25-$\mu$L reaction mixture contained 12.5 $\mu$L of 2× SYBR Premix Ex Taq II, 0.25 $\mu$L each of 10 $\mu$M forward and reverse primers, 0.5 $\mu$L of 50× ROXII reference dye, 9.5 $\mu$L of nuclease-free water, and 2 $\mu$L of cDNA template. The thermal cycling conditions were as follows: 1 cycle at 95°C for 30 s, followed by 40 cycles at 95°C for 5 s, 57°C for 30 s, and 72°C for 30 s. The primers and probes used are listed in Table 1. The 10 serially diluted plasmids containing the nuclear genes of the CVS-11 strain were used as standards for the RABV genomic RNA assay.

**Western blot analysis.** Whole-cell extracts were prepared in radioimmunoprecipitation assay (RIPA) buffer (Beyotime Biotechnology, China) containing 1 mM phenylmethylsulfonyl fluoride (PMSF; Beyotime Biotechnology, China). Protein concentrations in the cell lysates were quantified by the bicinchoninic acid (BCA) method (Pierce, IL, USA). Proteins were separated on sodium dodecyl sulfate-polyacrylamide gel electrophoresis (SDS-PAGE) gels and transferred onto 0.45-$\mu$m polyvinylidene fluoride (PVDF) membranes (Millipore, MA, USA). The membranes were blocked for 1 h at room temperature in Tris-buffered saline with Tween (TBST) containing 5% bovine serum albumin (BSA) and subsequently incubated with the indicated primary antibodies at 4°C overnight and then with the corresponding HRP-conjugated secondary antibodies at room temperature for 2 h. Signals were visualized on a Fujifilm LAS-4000 imager (Fujifilm, Tokyo, Japan) with a SuperSignal West Dura extended duration substrate kit (Pierce, IL, USA). Protein band densities were quantified using Multi Gauge software (version 3.1).

**Immunofluorescence assay.** N2a cells were seeded in laser confocal cell culture dishes and then transfected with the indicated plasmids or infected with CVS-11/treated with drugs. Accumulation of LC3 puncta was observed by transfection of green fluorescent protein (EGFP)-LC3. Cells were washed with PBS and fixed with 4% paraformaldehyde for 30 min. After permeabilization with 0.1% Triton X-100 for 10 min, cells were blocked with PBS containing 0.1% Tween 20 and 5% goat serum for 2 h. Cell nuclei were stained with 4′6-diamidino-2-phenylindole (DAPI; Sigma) for immunofluorescence assays. Lysosomes were stained with LysoTracker red. Fluorescence images were acquired using an Olympus confocal microscope (Olympus FV1000 confocal laser scanning microscope, Japan). Images were processed and analyzed using Olympus, ImageJ, and Photoshop (Adobe) software.

**Statistical analysis.** The results are expressed as the mean ± S.D. of three independent experiments. Unpaired Student's $t$ test was used to evaluate differences between the test samples and control samples. A $P$ value <0.05 was considered to indicate statistical significance. All statistical analyses and calculations were performed by using GraphPad Prism 5.

## ACKNOWLEDGMENTS

This work was supported by the National Natural Science Foundation of China (grant number 31872487).

We declare no conflicts of interest associated with this study.

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
