## [Reviewer comments · Microbiology Spectrum]

Microbiology Spectrum

Bif-1c attenuates viral proliferation by regulating autophagic flux blockade induced by the rabies virus CVS-11 strain in N2a cells

Pengfei Hou, Yidi Guo, Hongli Jin, Jingxuan Sun, Yujie Bai, Wujian Li, Ling Li, Zengguo Cao, Fangfang Wu, Haili Zhang, Yuanyuan Li, Songtao Yang, Xianzhu Xia, Pei Huang, and Hualei Wang

Corresponding Author(s): Hualei Wang, Jilin University

Review Timeline:

Submission Date:	August 6, 2022
Editorial Decision:	September 19, 2022
Revision Received:	November 18, 2022
Editorial Decision:	January 10, 2023
Revision Received:	January 17, 2023
Accepted:	March 9, 2023

Editor: Daniel Perez

Reviewer(s): The reviewers have opted to remain anonymous.

Transaction Report:

DOI: <https://doi.org/10.1128/spectrum.03079-22>

September 19, 2022

Prof. Hualei Wang
Jilin University
Key Laboratory of Zoonosis Research, Ministry of Education, College of Veterinary Medicine
Changchun
China

Re: Spectrum03079-22 (Bif-1c attenuates viral proliferation by regulating autophagic flux blockade induced by rabies virus CVS-11 strain in N2a cells)

Dear Prof. Hualei Wang:

Link Not Available

Sincerely,

Daniel Perez

Journals Department
Reviewer comments:

Reviewer #1 (Comments for the Author):

The manuscript entitled "Bif-1c attenuates viral proliferation by regulating autophagic flux blockade induced by rabies virus CVS-11 strain in N2a cells" by Hou et al. highlights an important role of neuron-specific/ubiquitous Bax-interacting factor-1 (Bif-1)-mediated autophagy in RABV proliferation. The authors found that the expression of Bif-1 was up-regulated by RABV infection. Overexpression of the isoform Bif-1c suppressed RABV infection through abolishing the autophagosome accumulation and prompting the autophagic flux induced by RABV. The study is interesting. However, there are some issues need to be addressed before publication:

Q1: To determine whether Bif-1 is differentially expressed in neuronal cells after RABV infection, N2a Cells were infected with RABV at a multiplicity of infection of 1. Further, do the authors have any evidence that the RABV with different MOI may affect Bif function?

Q2: Line in 292, which shRNA was used in the experiment, 1, 2, or 3? And could the shRNA for Bif-1 silence all the isoforms of Bif-1?

Q3: Please describe some details of the plasmids encoding Bif-1a, Bif-1b, Bif-1c, and Bif-1e, such as the vector. The name of plasmids and proteins should be distinguished in description. Thus, in line 459-460, authors should modify. And it may be better for accuracy to change the "Mock" to empty vector. There were no descriptions about the plasmids of DsRed1-Bif-1c, EGFP-Bif-1a, EGFP-Bif-1b, EGFP-Bif-1c, and EGFP-Bif-1e in reference 34.

Q4: In Fig.1, 2 and 4, if "hours post infection" has been annotated, "h" in after numbers could be deleted. And you could also use "h p.i." as an abbreviation for "hours post infection".

Q5: There are many grammar and spelling mistakes in the manuscript, the authors should check and correct them carefully.

Q6: There are many inconsistencies in the manuscript, please check carefully, such as "GFP-LC3" and "EGFP-LC3", "RT-qPCR" and "qRT-PCR", "μM" and "μm", "LC3-2" and "LC3-II"

Q7: The punctuation of article does not conform to the present specification, especially the space. Be correct and consistent with a space, or without a space.

Reviewer #2 (Comments for the Author):

In this manuscript, Hou and colleagues discover that Bax-interacting factor-1 (Bif-1) is upregulated upon rabies virus (RABV) infection. The overexpression of Bif-1 suppresses RABV replication and the knockdown of Bif-1 in turn promotes RABV replication. Furthermore, the authors clarify that the overexpression of Bif-1 promotes autophagosome fusion with lysosomes, therefore inhibits RABV replication.

The manuscript is well prepared and technically the study itself was performed with application of some new techniques. I have only several minor comments about this manuscript listed as below:

1. The effects of overexpression of several isoforms of Bif-1 on RABV replication are shown in Figure 3. Whether Bif-1a, Bif-1b, Bif-1c, and Bif-1e are successfully overexpressed is better verified by WB or indirect immunofluorescence assay (IFA).

2. The results in Figure 3 show that Bif-1a has no effect on RABV replication. According to the authors' hypothesis, this should be due to the fact that Bif-1a is not present in neurons in the natural environment, thus it cannot have an effect on autophagy process in neurons. I am more interested in whether Bif-1a can interact with LC3, this result can help us to confirm the important structural regions of Bif-1 involved in autophagy process in neurons. At 24 and 48 h.p.i., besides Bif-1c, other isoforms like Bif-1b and Bif-1e inhibit mRNA transcription of RABV, while there is no significant change of Bif-1a. How to explain this paradox?

3. Given that the experiments that infected CVS-11 in N2a and examined its effect on cellular autophagy have been published (Liu J et al., 2017, Autophagy), the reference to this article should be added to the text corresponding to the results in Figure 4.

4. Figures 5 and 6 demonstrate the effects of Bif-1c on incomplete autophagy induced by RABV infection, however the corresponding text does not elucidate why Bif-1c was chosen for the series of studies, but not Bif-1b or 1e. I speculate that Bif-1c is chosen because Bif-1c is the longest isoform of Bif-1, and its effect on RABV inhibition is the strongest among several isoforms of Bif-1. A similar description should better be added to the manuscript.

5. eGFP-LC3 and DsRed-Bif-1c were used for cell imaging experiments in Figure 5. Self-emitting fluorescent proteins are best imaged directly in the live cell state. The nucleus of live cells can be stained with Hoechst 33342. In short, I personally suggest that using live cells for imaging can achieve better imaging results, and the authors can improve the experimental protocol in their future studies.

6. The WB results shown in Figure 6A display two distinct bands for Bif-1, while Figure 1 shows three different bands. The authors should standardize the WB results shown in figures.

7. Similar to the comment 5, the confocal results in Figure 6 also used fixed cells followed by staining and imaging. Given the presence of lysosome's live cell dye, live cell imaging would provide better imaging results.

8. The authors should proof-read their manuscript carefully to avoid any typos and grammatical mistakes.

Staff Comments:

Preparing Revision Guidelines

Please return the manuscript within 60 days; if you cannot complete the modification within this time period, please contact me. If you do not wish to modify the manuscript and prefer to submit it to another journal, please notify me of your decision immediately so that the manuscript may be formally withdrawn from consideration by Microbiology Spectrum.

The manuscript entitled “Bif-1c attenuates viral proliferation by regulating autophagic flux blockade induced by rabies virus CVS-11 strain in N2a cells” by Hou et al. highlights an important role of neuron-specific/ubiquitous Bax-interacting factor-1 (Bif-1)-mediated autophagy in RABV proliferation. The authors found that the expression of Bif-1 was up-regulated by RABV infection. Overexpression of the isoform Bif-1c suppressed RABV infection through abolishing the autophagosome accumulation and prompting the autophagic flux induced by RABV. The study is interesting. However, there are some issues need to be addressed before publication:

Q1: To determine whether Bif-1 is differentially expressed in neuronal cells after RABV infection, N2a Cells were infected with RABV at a multiplicity of infection of 1. Further, do the authors have any evidence that the RABV with different MOI may affect Bif function?

Q2: Line in 292, which shRNA was used in the experiment, 1, 2, or 3? And could the shRNA for Bif-1 silence all the isoforms of Bif-1?

Q3: Please descript some details of the plasmids encoding Bif-1a, Bif-1b, Bif-1c, and Bif-1e, such as the vector. The name of plasmids and proteins should be distinguished in description. Thus, in line 459-460, authors should modify. And it may be better for accuracy to change the “Mock” to empty vector. There were no descriptions about the plasmids of DsRed1-Bif-1c, EGFP-Bif-1a, EGFP-Bif-1b, EGFP-Bif-1c, and EGFP-Bif-1e in reference 34.

Q4: In Fig.1, 2 and 4, if “hours post infection” has been annotated, “h” in after numbers could be deleted. And you could also use “h p.i.” as an abbreviation for “hours post infection”.

Q5: There are many grammar and spelling mistakes in the manuscript, the authors should check and correct them carefully.

Q6: There are many inconsistencies in the manuscript, please check carefully, such as “GFP-LC3” and “EGFP-LC3”, “RT-qPCR” and “qRT-PCR”, “ μ M” and “ μ m”, “LC3-2” and “LC3-II”

Q7: The punctuation of article does not conform to the present specification, especially the space. Be correct and consistent with a space, or without a space.

Dear Editor,

Thank you for your decision letter concerning our manuscript entitled “Bif-1c attenuates viral proliferation by regulating autophagic flux blockade induced by the rabies virus CVS-11 strain in N2a cells” and for your time regarding our revision. I also appreciate all of the constructive comments from you and the reviewers. We have revised the manuscript according to these comments. With these improvements, we hope that the current version meets the journal’s standards for publication. The following are point-by-point responses to all of those comments and a list of changes that we have made to the manuscript.

Sincerely,
Prof. Hualei Wang

Point-by-point responses to the comments of the Editor and reviewers, and a list of changes:

Reviewer #1 (Comments for the Author):

The manuscript entitled "Bif-1c attenuates viral proliferation by regulating autophagic flux blockade induced by rabies virus CVS-11 strain in N2a cells" by Hou et al. highlights an important role of neuron-specific/ubiquitous Bax-interacting factor-1 (Bif-1)-mediated autophagy in RABV proliferation. The authors found that the expression of Bif-1 was up-regulated by RABV infection. Overexpression of the isoform Bif-1c suppressed RABV infection through abolishing the autophagosome accumulation and prompting the autophagic flux induced by RABV. The study is interesting. However, there are some issues need to be addressed before publication:

Q1: To determine whether Bif-1 is differentially expressed in neuronal cells after RABV infection, N2a Cells were infected with RABV at a multiplicity of infection of 1. Further, do the authors have any evidence that the RABV with different MOI may affect Bif function?

A1: That is a very good question. Bif-1c-overexpressing N2a cells were infected with CVS-11 at an MOI of 1 in the original experiment. To further validate the role of Bif-1c, N2a cells were also infected with CVS-11 at an MOI of 0.5 and 2 after empty vector or Bif-1c transfection. Consistent with previous data obtained after infection at an MOI of 1, Bif-1c overexpression decreased the mRNA level of RABV N (Figure R1, shown below), suggesting that the MOI had no significant effect on the role of Bif-1c during CVS-11 infection.

Fig. R1 Overexpression of Bif-1c attenuates RABV infection in N2a cells. N2a cells transfected with Bif-1c or empty vector were infected with CVS-11 at an MOI of 0.5 and 2. The RNA copy numbers of RABV N were determined by qRT-PCR analysis at 24 h p.i.

Q2: Line in 292, which shRNA was used in the experiment, 1, 2, or 3? And could the shRNA for Bif-1 silence all the isoforms of Bif-1?

A2: That is a very good question. Western blot was used to compare the silencing efficiency of the three shRNAs after transfection, as shown in Fig. R2. shRNA1 was the most efficient of the three shRNAs in knocking down Bif-1. To avoid confusing the reader, the description of shRNA2 and shRNA 3 has been removed. The shRNA used to knock down Bif-1 could silence all Bif-1 isoforms. The related information has been added to lines 311-313: “shRNA targeting Bif-1 (shBif-1: 5’-AGGAATTGAGAATAACTCAAAG-3’) and negative control shRNA (shNC: 5’-GTTCTCCGAACGTGTCACGT-3’) were purchased from GenePharma.”

Fig. R2 The verification of shRNA efficiency to silence Bif-1. Three shRNAs were transfected into N2a cells, and the cells were harvested for further analysis after 36 h.

Q3: Please describe some details of the plasmids encoding Bif-1a, Bif-1b, Bif-1c, and Bif-1e, such as the vector. The name of plasmids and proteins should be distinguished in description. Thus, in line 459-460, authors should modify. And it may be better for accuracy to change the "Mock" to empty vector. There were no descriptions about the plasmids of DsRed1-Bif-1c, EGFP-Bif-1a, EGFP-Bif-1b, EGFP-Bif-1c, and EGFP-Bif-1e in reference 34.

A3: Thank you for these constructive suggestions. We have supplemented the details of the constructed plasmids in **lines 289-306**: “The genes encoding the mouse Bif-1a and Bif-1e isoforms were PCR-amplified from N2a cell genomic cDNA with gene-specific primers (upstream primer 5’ -ATGAACATCATGGATTTC AACGTG-3’ and downstream primer 5’ -TTAATTGAGAAGTTCTAAGTAGGTAATTGG-3’) and cloned into the pIRES2-EGFP vector. The Bif-1b gene was amplified from N2a cell genomic cDNA using 2 pairs of specific primers (pair 1: upstream primer 5’-ATGAACATCATGGATTTC AACGTG-3’ and downstream primer: 5’-TTTTGTCACTTCTCTGCCCAAATCATAATG-3’ ; pair 2: upstream primer 5’ -ATTTGGGCAGAGGAAGTGACAAAATCTG-3’ and downstream primer 5’-TTAATTGAGAAGTTCTAAGTAGGTAATTGG-3’) and cloned into the pIRES2-EGFP vector. The Bif-1c isoform was amplified with 2 pairs of specific primers (pair 1: upstream primer: 5’-ATGAACATCATGGATTTC AACGTG-3’ and downstream primer 5’-TACATGCAGGAAGTTGAGCATGTAAGAGAAATTTACC ATAATGTTATCTCCTTCAG-3’; pair 2: upstream primer 5’-TACATGCTCAACTTCTGCATGTAAAATGGCTGAAGATTTGGGCAGAGG AAGTG-3’ and 5’-TTAATTGAGAAGTTCTAAGTAGGTAATTGG-3’) for the pIRES2-EGFP vector and pDsRed1-N1 vector.” “Mock” has been replaced with “empty vector (EV)” in Fig. 3 and the corresponding legend.

Q4: In Fig.1, 2 and 4, if "hours post infection" has been annotated, "h" in after numbers could be deleted. And you could also use "h p.i." as an abbreviation for "hours post infection".

A4: Thank you for pointing this out. This mistake has been revised in Figs. 1, 2 and 4.

Q5: There are many grammar and spelling mistakes in the manuscript, the authors should check and correct them carefully.

A5: Thank you for pointing this out. We have checked the entire manuscript, and the mistakes have been corrected.

Q6: There are many inconsistencies in the manuscript, please check carefully, such as "GFP-LC3" and "EGFP-LC3", "RT-qPCR" and "qRT-PCR", "μM" and "μm", "LC3-2" and "LC3-II"

A6: Thank you for pointing this out. We have checked the entire manuscript and corrected all of these inconsistencies.

Q7: The punctuation of article does not conform to the present specification, especially the space. Be correct and consistent with a space, or without a space.

A7: We thank the reviewer for this suggestion to improve our manuscript. We have checked and modified the punctuation and formatting of the manuscript again. All changes are highlighted in the new version of the manuscript.

Reviewer #2 (Comments for the Author):

In this manuscript, Hou and colleagues discover that Bax-interacting factor-1 (Bif-1) is upregulated upon rabies virus (RABV) infection. The overexpression of Bif-1 suppresses RABV replication and the knockdown of Bif-1 in turn promotes RABV replication. Furthermore, the authors clarify that the overexpression of Bif-1 promotes autophagosome fusion with lysosomes, therefore inhibits RABV replication. The manuscript is well prepared and technically the study itself was performed with application of some new techniques. I have only several minor comments about this manuscript listed as below:

Q1: The effects of overexpression of several isoforms of Bif-1 on RABV replication are shown in Figure 3. Whether Bif-1a, Bif-1b, Bif-1c, and Bif-1e are successfully overexpressed is better verified by WB or indirect immunofluorescence assay (IFA).

A1: Thank you for the constructive suggestions. The seven following transcript variants that can be transcribed into Bif-1a, Bif-1b, Bif-1c, and Bif-1e were obtained by PCR and overlap PCR from N2a cell mRNA: transcript variant V1 (Bif-1a, NM_019464. 3), transcript variant V2 (Bif-1a, NM_001282042. 1), transcript variant V3 (Bif-1b, NM_001282037. 1), transcript variant X1 (Bif-1c, XM_006501716. 3), transcript variant X2 (Bif-1c, XM_006501717. 1), transcript variant X3 (Bif-1c, XM_006501718. 3) and Bif-1e (unreported variant). Sequence analysis of seven Bif-1 alternatively spliced isoforms is shown in Fig. R3a. In the original manuscript, the transcript sequences V2, V1, X3 and Bif-1e were chosen to explore the effect of overexpression of the four Bif-1a, Bif-1b, Bif-1c and Bif-1e isoforms on RABV replication. The PCR products were cloned into the pIRES2-EGFP vector to construct the recombinant expression plasmids. N2a cells were transfected with the seven recombinant expression plasmids and empty vector, respectively, and green fluorescence was visible under a fluorescence microscope after 24 h of transfection, except for the cells transfected with empty vector (Fig. R3b). The expression of Bif-1a, Bif-1b, Bif-1c, and Bif-1e was verified by western blot (Fig. R3c). The relevant description is provided **in lines 134-138**: “N2a cells were transfected with recombinant plasmids encoding Bif-1a, Bif-1b, Bif-1c, Bif-1e and EV, respectively. After 36 h, cell lysates were collected and analyzed by western blot. Bands

corresponding to the expected sizes of Bif-1a, Bif-1b, Bif-1c, and Bif-1e were detected, indicating successful expression of the target proteins (data not shown).”

Fig. R3 Overexpression of Bif-1a, Bif-1b, Bif-1c, and Bif-1e in N2a cells was verified. (A) Sequence comparative analysis of seven Bif-1 alternatively spliced isoforms. Transcript variant V1 (V1, NM_019464. 3), transcript variant V2 (V2, NM_001282042. 1), transcript variant V3 (V3, NM_001282037. 1), transcript variant X1 (X1, XM_006501716. 3), transcript variant X2 (X2, XM_006501717. 1), transcript variant X3 (X3, XM_006501718. 3), Bif-1e (unreported variant). (B) Fluorescence analysis of N2a cells transfected with recombinant overexpression plasmids ($\times 200$). (C) Overexpression of Bif-1a, Bif-1b, Bif-1c, and Bif-1e in N2a cells was verified by western blot. N2a cells were transfected with recombinant plasmids encoding the CDS regions of the different Bif-1 transcript variants. Equal amounts of cell lysates from each preparation were analyzed. The indicated plasmids were selected for downstream experiments.

Q2: The results in Figure 3 show that Bif-1a has no effect on RABV replication. According to the authors' hypothesis, this should be due to the fact that Bif-1a is not present in neurons in the natural environment, thus it cannot have an effect on autophagy process in neurons. I am more interested in whether Bif-1a can interact with LC3, this result can help us to confirm the important structural regions of Bif-1 involved in autophagy process in neurons. At 24 and 48 h.p.i., besides Bif-1c, other isoforms like Bif-1b and Bif-1e inhibit mRNA transcription of RABV, while there is no significant change of Bif-1a. How to explain this paradox?

A2: Thank you for your constructive suggestions. Early in the study, we simultaneously constructed the pDsRed1-Bif-1a with pDsRed1-Bif-1c plasmids. After

RABV infection for 36 h, the distribution of DsRed1-Bif-1a and EGFP-LC3 was observed by confocal microscope. The results showed that DsRed1-Bif-1a and EGFP-LC3 were diffusely distributed throughout the cytoplasm of N2a cells under RABV infected, with limited colocalization (Fig. R4). In comparison, Bif-1c was observed to co-localize more with LC3 in the cytoplasm (Fig. 5). It is crucial to determine whether the Bif-1 isoform can interact with LC3 to help us further explore the mechanism by which Bif-1 is involved in the autophagy process in neurons. We have been working on determining this information. Moreover, we also attempted to confirm the interactions between Bif-1 and RABV proteins. Bif-1c had a greater effect than Bif-1b or Bif-1e on RABV replication, and we speculated that this was because Bif-1c contains the most integrated functional domain.

Fig. R4 Intracellular localization of Bif-1a and LC3. N2a cells were transfected with EGFP-LC3 and pDsRed1-Bif-1a and then infected with CVS-11 (MOI: 5). At 48 h p.i., the cells were fixed and visualized by confocal microscopy. The intracellular colocalization of Bif-1 and LC3 puncta is indicated by arrows. DAPI (blue) was used to stain nuclear DNA. Scale bar, 10 µm. EV, empty vector.

Q3: Given that the experiments that infected CVS-11 in N2a and examined its effect on cellular autophagy have been published (Liu J et al., 2017, Autophagy), the reference to this article should be added to the text corresponding to the results in Figure 4.

A3: Thank you for pointing this out. This reference has been added to the sentence in **line 163**. We now discuss this in the Discussion section.

Q4: Figures 5 and 6 demonstrate the effects of Bif-1c on incomplete autophagy induced by RABV infection, however the corresponding text does not elucidate why Bif-1c was chosen for the series of studies, but not Bif-1b or 1e. I speculate that Bif-1c is chosen because Bif-1c is the longest isoform of Bif-1, and its effect on RABV inhibition is the strongest among several isoforms of Bif-1. A similar description should better be added to the manuscript.

A4: Thank you for your constructive suggestions. We chose to examine the function of Bif-1c in RABV-induced autophagy because Bif-1c contains the most integrated functional domains and attenuated viral replication to the greatest extent. We have discussed this in the Discussion section. A description has also been added to the Results in lines 166-169: “Considering the crucial roles of Bif-1 in regulating cell autophagy and because the neuron-specific Bif-1 isoform Bif-1c had the greatest effect on RABV replication, we next examined the mechanism by which Bif-1c functions in RABV-induced autophagy.”

Q5: eGFP-LC3 and DsRed-Bif-1c were used for cell imaging experiments in Figure 5. Self-emitting fluorescent proteins are best imaged directly in the live cell state. The nucleus of live cells can be stained with Hoechst 33342. In short, I personally suggest that using live cells for imaging can achieve better imaging results, and the authors can improve the experimental protocol in their future studies.

A5: Thank you for your constructive suggestions. It’s a great way to achieve better imaging results by using self-emitting fluorescent proteins in live cell and to ascertain our conclusion. We will improve the experimental protocol in future studies.

Q6: The WB results shown in Figure 6A display two distinct bands for Bif-1, while Figure 1 shows three different bands. The authors should standardize the WB results shown in figures.

A6: Thank you for pointing this out. We apologize for not showing this result clearly, in fact in the unprocessed figure shows three bands for Bif-1 (Fig. R5). In the original manuscript, we cropped figure of the western blot and were shown due to considering the scale of figures in Fig. 6A. To avoid confusing the reader, we have uploaded the complete figure to replace the original Fig. 6.

Fig. R5 N2a cells were transfected with Bif-1c (transcript variant X3) and then infected with CVS-11 (MOI: 0.1) or mock-treated. After 48 h, cell lysates were collected, and the expression levels of Bif-1 were determined by western blot analysis.

Q7: Similar to the comment 5, the confocal results in Figure 6 also used fixed cells followed by staining and imaging. Given the presence of lysosome's live cell dye, live cell imaging would provide better imaging results.

A7: Thank you for your constructive suggestions. We have not clearly explained that we indeed employed live-cell dye in Fig. 6. Fluorescence images were acquired after lysosomes were stained with LysoTracker Red. The live-cell staining content has been added to the Materials and Methods. We added a description of this procedure **in lines 189-191**: “We transfected N2a cells with EGFP-LC3 and labeled lysosomes with LysoTracker Red (50 nM), and live-cell imaging was performed.”

Q8: The authors should proof-read their manuscript carefully to avoid any typos and grammatical mistakes.

A8: Thank you for pointing this out. We have checked the entire manuscript, and mistakes have been corrected.

January 10, 2023

Prof. Hualei Wang
Jilin University
Key Laboratory of Zoonosis Research, Ministry of Education, College of Veterinary Medicine
Changchun
China

Re: Spectrum03079-22R1 (Bif-1c attenuates viral proliferation by regulating autophagic flux blockade induced by the rabies virus CVS-11 strain in N2a cells)

Dear Prof. Hualei Wang:

Link Not Available

Sincerely,

Daniel Perez

Journals Department
Reviewer comments:

Reviewer #1 (Comments for the Author):

In this paper, the authors illustrated the associations between Bif-1 and RABV infection. The expression of Bif-1 was up-regulated by RABV infection. Overexpression of the isoform Bif-1c suppressed RABV infection. Research data indicated that neuron-specific Bif-1 isoforms impair the replication process of RABV by abolishing autophagosome accumulation and blocking autophagic flux induced by RABV in mouse neuroblastoma (N2a) cells. The article is quite innovative, and its conclusion is reliable.

Reviewer #2 (Comments for the Author):

In this revised version, all of the points raised by the reviewers have been answered properly. Some new results and discussion have been added in this version. The manuscript is thus greatly improved and can be considered for acceptance.

But a couple of mistakes need to be fixed before publication:

- (1). In line 3, "abstract" is unnecessary.
- (2) The table 1 is a little bit confused and seems that the probe for RABV-N is missing.
- (3) The scare bars for IFA in Fig 4 and 6 need to be described in the figure legends.

Staff Comments:

Preparing Revision Guidelines

Please return the manuscript within 60 days; if you cannot complete the modification within this time period, please contact me. If you do not wish to modify the manuscript and prefer to submit it to another journal, please notify me of your decision immediately so that the manuscript may be formally withdrawn from consideration by Microbiology Spectrum.

In this paper, the authors illustrated the associations between Bif-1 and RABV infection. The expression of Bif-1 was up-regulated by RABV infection. Overexpression of the isoform Bif-1c suppressed RABV infection. Research data indicated that neuron-specific Bif-1 isoforms impair the replication process of RABV by abolishing autophagosome accumulation and blocking autophagic flux induced by RABV in mouse neuroblastoma (N2a) cells. The article is quite innovative, and its conclusion is reliable. It would provide a new perspective on understanding host-virus interaction. Overall, the revised manuscript is a clearer, smoother version of the original. It is easy to comprehend.

Reviewer #1 (Comments for the Author):

In this paper, the authors illustrated the associations between Bif-1 and RABV infection. The expression of Bif-1 was up-regulated by RABV infection. Overexpression of the isoform Bif-1c suppressed RABV infection. Research data indicated that neuron-specific Bif-1 isoforms impair the replication process of RABV by abolishing autophagosome accumulation and blocking autophagic flux induced by RABV in mouse neuroblastoma (N2a) cells. The article is quite innovative, and its conclusion is reliable.

Response: Thanks a lot for your helps on this paper.

Reviewer #2 (Comments for the Author):

In this revised version, all of the points raised by the reviewers have been answered properly. Some new results and discussion have been added in this version. The manuscript is thus greatly improved and can be considered for acceptance.

But a couple of mistakes need to be fixed before publication:

(1) In line 3, "abstract" is unnecessary.

Response: We apologize for the mistakes, and we have also deleted "abstract" from line 3 of the original manuscript.

(2) The table 1 is a little bit confused and seems that the probe for RABV-N is missing.

Response: Thank you for pointing this out. We have rearranged the layout of Table 1. In addition, since SYBR Green real-time PCR was used for the RABV *N* gene, we used only forward and reverse primers without probe.

(3) The scale bars for IFA in Fig 4 and 6 need to be described in the figure legends.

Response: Thank you for pointing this out. We have supplemented the description of scale bars for IFA in Fig 4 and 6: "Bars, 10 μ m".

March 9, 2023

Prof. Hualei Wang
Jilin University
Key Laboratory of Zoonosis Research, Ministry of Education, College of Veterinary Medicine
Changchun
China

Re: Spectrum03079-22R2 (Bif-1c attenuates viral proliferation by regulating autophagic flux blockade induced by the rabies virus CVS-11 strain in N2a cells)

Dear Prof. Hualei Wang:

Your manuscript has been accepted, and I am forwarding it to the ASM Journals Department for publication. You will be notified when your proofs are ready to be viewed. I apologize for the delay as I am dealing with personal issues.

Sincerely,

Daniel Perez
Editor, Microbiology Spectrum
